# Wealth and obesity in pre-adolescents and their guardians: A first step in explaining non-communicable disease-related behaviour in two areas of Nairobi City County

**Sophie Ochola**[1]*, **Noora Kanerva**[2], **Lucy Joy Wachira**[3], **George E. Owino**[4], **Esther L. Anono**[1], **Hanna M. Walsh**[2], **Victor Okoth**[5], **Maijaliisa Erkkola**[2], **Nils Swindell**[6], **Gareth Stratton**[6], **Vincent Onywera**[3], **Mikael Fogelholm**[2]

1 Department of Food, Nutrition and Dietetics, Kenyatta University, Nairobi, Kenya, 2 Department of Food and Nutrition, University of Helsinki, Helsinki, Finland, 3 Department of Physical Education, Exercise and Sport Science, Kenyatta University, Nairobi, Kenya, 4 Department of Sociology, Gender and Development Studies, Kenyatta University, Nairobi, Kenya, 5 Department of Environmental Science, Kenyatta University, Nairobi, Kenya, 6 Faculty of Science and Engineering, Applied Sport Technology, Exercise and Medicine Research Centre, Swansea University, Swansea, United Kingdom

* ochola.sophie@ku.ac.ke

**Data Availability Statement:** The dataset has been submitted to Dryad (https://datadryad.org/stash) and the citation is: Kanerva, Noora (2022), KENFIN-

## Abstract

The prevalence of non-communicable diseases is increasing in lower-middle-income countries as these countries transition to unhealthy lifestyles. The transition is mostly predominant in urban areas. We assessed the association between wealth and obesity in two sub-counties in Nairobi City County, Kenya, in the context of family and poverty. This cross-sectional study was conducted among of 9–14 years old pre-adolescents and their guardians living in low- (Embakasi) and middle-income (Langata) sub-counties. The sociodemographic characteristics were collected using a validated questionnaire. Weight, height, mid-upper arm circumference, and waist circumference were measured using standard approved protocols. Socioeconomic characteristics of the residential sites were accessed using Wealth Index, created by using Principal Component Analysis. Statistical analyses were done by analysis of variance (continuous variables, comparison of areas) and with logistic and linear regression models.A total of 149 households, response rate of 93%, participated, 72 from Embakasi and 77 from Langata. Most of the participants residing in Embakasi belonged to the lower income and education groups whereas participants residing in Langata belonged to the higher income and education groups. About 30% of the pre-adolescent participants in Langata were overweight, compared to 6% in Embakasi (p<0.001). In contrast, the prevalence of adults (mostly mothers) with overweight and obesity was high (65%) in both study areas. Wealth (β = 0.01; SE 0.0; p = 0.003) and income (β = 0.29; SE 0.11; p = 0.009) predicted higher BMI z-score in pre-adolescents. In, pre-adolescent overweight was already highly prevalent in the middle-income area, while the proportion of women with overweight/obesity was high in the low-income area. These results suggest that a lifestyle promoting obesity is high regardless of socioeconomic status and wealth in

EDURA, Dryad, Dataset, https://doi.org/10.5061/dryad.2jm63xssn.

**Funding:** This study was part of a collaborative project "The Kenya-Finland Education and Research Alliance (KENFIN-EDURA)" (state grant HEL7M0453-82, sum 527,000 EUR awarded to MF. and VO) funded by the Finnish Ministry of Foreign Affairs through The Higher Education Institutions Institutional Cooperation Instrument (HEI ICI; https://www.oph.fi/en/programmes/hei-ici-programme). The funders had no role in study design, data collection and analysis, decision to publish, or preparation of the manuscript.

**Competing interests:** The authors have declared that no competing interests exist.

**Abbreviations:** DHS, Demographic Health Surveys; FFQ, Food Frequency Questionnaire; KMO, Kaiser-Meyer-Olkin; KUERC, Kenyatta University Ethical Review Committee; MUAC, Middle Upper Arm Circumference; NCDs, Non Communicable Diseases; ODK, Open Data Kit; PCA, Principal Component Analysis; SES, Socioeconomic Status; SPSS, IBM Statistical Package for Social Sciences; WC, Waist Circumference; WHO, World Health Organization.

Kenya. This provides a strong justification for promoting healthy lifestyles across all socio-economic classes.

## Introduction

The contribution of non-communicable diseases (NCDs) to morbidity and mortality in lower middle-income countries (LMIC) is continuously increasing and at a faster rate than in high-income countries [1]. Hence, LMICs are facing a triple burden of NCDs, communicable diseases and undernutrition. In LMICs, the share for communicable diseases and infections was 27% among the top 10 causes of death in 2015 and for NCDs the share was 63% [2] whereas in upper-middle income and high-income countries, NCDs explained already more than 90% of deaths related to top 10 causes.

Kenya has been recently classified as a LMIC by the World Bank [3] and it is undergoing a very rapid lifestyle transition. The existing data collected mainly before 2010 indicates that the country is reaching the point where unhealthy lifestyle that follows economic development and increase the risk of chronic diseases start to entrench unwanted health gaps between genders (more females) [4,5] and between living environments (more in urban than rural areas) [6,7]. In 2014, overweight and obesity among women was 33% in total and positively related to both education and wealth [8]. Inadequate physical activity was more frequent among women than among men, it was high also in those with no education, but low among poor people [9]. In Nairobi City County, 21% of children were overweight or obese, with higher rates among girls and those of higher socioeconomic status [10,11]. Fast and strong actions are needed to at least slow down the increasing prevalence of NCDs and widening of inequalities in Kenya. To plan better health policies and health promotion actions, a step also in science should be taken from describing the situation to better understanding of determinants of current health behaviour and how it could be changed.

The present paper is the first from a multidisciplinary research project aimed at collecting novel and important in-depth data on determinants of NCD-related lifestyles and risk factors among families (pre-adolescents and their guardians) residing in Nairobi City, Kenya, with special focus on comparing two socioeconomically different areas.

The overall aim of the study was to get insight on the role of poverty, education, gender, and place of residence as determinants of physical (in)activity, dietary quality and variability, and overweight/obesity. This opening paper analyses the associations between indicators of wealth and obesity in guardians and pre-adolescent children in a low- and middle-income areas in urban Nairobi and includes a description of the overall study protocol.

## Methods

### Study design

This cross-sectional study was part of a 3-year collaborative project "The Kenya-Finland Education and Research Alliance (KENFIN-EDURA)" funded by the Finnish Ministry of Foreign Affairs through the Higher Education Institutions Institutional Cooperation Instrument (HEI ICI). The aim of this project was to improve education and teaching capacity at the Kenyatta University, Nairobi, with a focus on NCDs-related behavior, mainly diet and physical activity. The overall study was multi-disciplinary and it included several tasks. In this paper, we report methods and results from the socio-demographic indices, including wealth and anthropometric assessments. The study design in shown in S1 Text.

## Study sites

The study was carried out in Embakasi Central sub-county (Kayole South Ward) and Langata sub-county (Nairobi West Ward) within Nairobi County. When referring to these areas, we use "Embakasi" and "Langata". These areas were selected since people who reside there are from low to middle socioeconomic status (SES) [12]. People in these SES groups are likely to be the most affected by a recent transition in health behaviour (towards unhealthy diet and inadequate physical activity) [12,13]. Furthermore, the pre-adolescents and their families living in these sub-counties have not been previously studied in such detail. The map of the study sites is shown in S1 Text.

## Study population and sample size

The study targeted families of low or middle SES with pre-adolescents in the age range of 9–14 years and their guardian(s), residing in the selected study regions in Nairobi County. The main inclusion criteria were that the family has at least one child aged from 9 to 14 years and at least one parent or guardian available, and who have been residents of Langata or Embakasi for at least 6 months before the study. Moreover, the family had to sign an informed consent, hence be a voluntarily participant. The study included the pre-adolescent and guardian(s) (mother and/or father, if both belonged to the family and regardless of whether they were biological parents or not). If there was more than one pre-adolescent in the target age range, the participating child was drawn randomly. Households with a 9–14 years old with documented chronic disease conditions, such as tuberculosis, impacting diet or who had any significant illness preventing participation were excluded from the study.

The quantitative part of the study was mainly descriptive, and the study had multiple outcome measures. However, one main outcome was childhood overweight/obesity which was used as the basis for power calculation. According to Broyles et al. [14], an expected difference in child BMI z-score between low- and middle-income SES in low-income countries is 0.5. When we use 1.0 as the SD, and $\alpha = 0.05$ and power = 80%, a total 2-group sample size is n = 126 (n = 63 per group). To allow for non-response, we aimed to invite in total 160 households, i.e., 80 from each for the study area.

## Sampling technique

A multi-stage sampling technique was conducted to identify the households where data would be collected (Fig 3 in S1 Text). Embakasi represented low SES (partly an informal settlement) and Langata the middle SES [15]. Next, five villages were selected randomly from Kayole North Central Ward (from Embakasi) and 12 estates from Nairobi West Ward (from Langata). These wards were purposively selected from the sub-counties, since they are densely populated and would therefore provide adequate sample of the target population (pre-adolescents). The final stage of sampling involved the enumeration of households with a child in the age range 9–14 years from the selected villages and estates with the assistance of Community Health Volunteers (CHVs). In total 223 households were enumerated in Embakasi and 173 households in Langata. Simple random sampling technique was used to select 80 households from both sub-counties. The final sample sizes were 72 for Embakasi and 77 for Langata (S1 Text). There were only few families (8 in Embakasi and 3 in Langata) that did not want to take part and no data was collected from them.

## Data collection and processing

Field assistants visited the households twice approximately 8 days apart during April-June 2019. All data analysed in this paper were collected during the first visit: the pre-adolescents

and guardians completed questionnaires, including questions on demographic and socioeconomic background. Moreover, the participants' anthropometric measurements were taken. For each household, the assessments took in total about 4 h, divided on two occasions. The time-consuming and multiple assessments were a restriction for a much larger study sample.

## Socio-demographic characteristics

The interviewer-administered questionnaires were done in English or Kiswahili languages, depending on the preference of the family. The quantitative data was collected electronically by use of android phones/tablets using purposefully developed questionnaires developed on Open Data Kit (ODK) software [16], in a face-to-face interview during household visits.

A structured and validated questionnaire was used to collect demographic and socioeconomic characteristics, as well as housing conditions and ownership of assets [8]. This questionnaire was administered to guardians. The questionnaire included information on participants' sex, age, marital status, education, income, occupation, form of employment, household expenditure on various items, the materials from which the household is constructed based on the questionnaire used in the Demographic and Health Survey in Kenya [8]. Wealth is often measured in terms of economic status and living standards of households. However, the income, expenditure and consumption data needed for calculating these can be challenging to measure accurately. For this reason, we chose to use the Wealth Index similarly to the Demographic and Health Surveys (DHS) [8] and World Food Programme (WFP) Surveys [17]. The calculation of the Index is explained in the statistical analyses section.

## Anthropometric measurements

Weight and height of pre-adolescents and their guardians were measured with minimal clothing on and shoes off, using a digital electronic scale (Seca Robust 813) to the nearest 0.1 kg and a stadiometer (Seca 217) to the nearest 0.1 cm. BMI-for-age z-scores for pre-adolescents were calculated, using WHO's growth references [18]. For adults, BMI (kg/m2) was calculated by dividing weight for the square of height. Waist circumference (WC) was measured, using a waist circumference tape (Edtape for body measurements) to the nearest 0.1 cm around one's body about halfway between the bottom of the lowest rib and the top of the hip bone, roughly in line with their belly button over light clothing or skin. The Mid-Upper Arm Circumference (MUAC) of the left upper arm was measured at the mid-point between the tip of the shoulder and the tip of the elbow using a non-elastic anthropometric measuring tape (to the nearest 0.1 cm). Each of the measurements was taken twice and an average calculated to ensure accuracy. For pre-adolescents, underweight was defined as BMI z-score < -2.0, overweight as BMI z-score >1.0 and obesity as BMI z-score >2.0. For thinness, we used MUAC values <18.5 cm and <16.0 cm, the latter indicating severe thinness. For adult underweight, overweight and obesity, we used international cut-offs of BMI <18.5 kg/m2, >24.9 kg/m2, and >29.9 kg/m2. Waist/abdominal obesity was defined as >88.0 cm for women and >102 cm for men. For adults, MUAC cut-offs for underweight were defined as <22.0 cm for women and <23.cm for men.

## Statistical analysis

Statistical analyses were done using SPSS (IBM SPSS Statistics version 25) and R software (version 3.6.3 for mac) [19]. Household characteristics were calculated for the total population and by study area as means and standard errors for continuous variables and as counts and percentages for categorical variables. Difference in anthropometric measurements between study areas was analyzed using analysis of variance or logistic regression. The continuous

anthropometric measures (BMI, z-BMI score, MUAC and waist circumference) were treated as outcome variables and a binary variable indicating study area was used as independent variable in the analysis of variance. In the logistic regression, the outcome variables were binary variables for overweight (for adults coded as BMI >24.9 = 1, others = 0; for adolescents coded as BMI z-score >1.0 = 1, others = 0), obesity (for adults, coded as BMI >29.9 = 1, others = 0; for adolescents coded as BMI z-score >2.0 = 1, others = 0) and central obesity (coded as WC >88cm for women and WC>102cm for men = 1, others = 0). All analyses were adjusted of participants' sex and age. Further, we analyzed the associations of wealth (categorical independent variable) and income (categorical independent variable) with BMI and BMI z-score (continuous outcome variables) by using a linear regression model, adjusted for age and sex. The same analyses with obesity and overweight as binary outcome variables were done by using a logistic regression model.

The Wealth Index was created according to the WFP VAM Guidance Paper [20]. Principal component analysis (PCA) (SPSS command FACTOR, Method: Principal components, Varimax-rotation) was applied to combine information on asset ownership and housing characteristics. Where necessary, variables were recoded into binary variables based on knowledge and insight of Kenyan researchers. Additional information related to the construction of the Wealth Index are presented in S2 Text.

First, the frequencies of the wealth indicators were explored in both areas. The indicators that existed in over 95% or less than 5% of the households were removed. These included: electricity, mobile phone, solar panel, table, sofa and bed. The Kaiser-Meyer-Olkin (KMO) test was used to determine the sampling adequacy of data and to ensure that the data were suitable to run a Factor Analysis. KMO values between 0.8 and 1 were deemed to indicate adequate sampling. The correlation test printed out with the PCA was used to evaluate whether correlations were too high meaning certain variables measure the same thing. This led to removing improved drinking water, motorcycle, improved wall material, radio, and cassette or CD player. The final wealth index had a KMO value of 0.871 and it explained 40% of the total variation (S2 Text). The index included sanitation, floor material, television, refrigerator, chair, cupboard, wall clock, microwave, DVD player, electric or gas stove, kerosene stove, bicycle and car or truck. The wealth index was grouped by quintile classification.

### Ethical considerations

Ethical clearance was sought from the Kenyatta University Ethical Review Committee (KUERC) and a research permit from the National Commission for Science, Technology and Innovation (Ethical clearance number: PKU/946/I1002). Further clearance was sought from the Sub-County Health Management Teams (from Embakasi and Langata) and from the local administration (chiefs) for the areas where the study was to be conducted before its commencement. Eligible pre-adolescents along with their guardian(s) were given an informed consent form to sign to show willingness to participate in the study. The purpose of the study, the interviews to be done, the voluntary nature of participation, and the right to refuse to participate in any part of the study were also to be explained orally.

### Results

Altogether, 149 households (93% of those invited) participated in the study (Table 1). Seventy-two (72) families participated in Embakasi and 77 in Langata. Of the participating guardians, almost all were women, whereas from the 9–14 years old pre-adolescents, about half were girls. The pre-adolescents' mean age (11 years) was similar, but guardians were younger in Embakasi compared to Langata (32 vs. 38 years). In both areas, the average household size was

**Table 1. Sociodemographic characteristics of participants (9–14 years old pre-adolescents [n = 149] and their guardians [n = 148]) in Embakasi and Langata, Nairobi.**

| | Embakasi | | Langata | | Total | |
|---|---|---|---|---|---|---|
| | Mean (SD) or N | % | Mean (SD) or N | % | Mean (SD) or N | % |
| Number of households | 72 | | 77 | | 149 | |
| Household size | 4.9 (1.4) | | 5.1 (1.6) | | 5 (1.5) | |
| **All guardians** | 72 | 100.0 | 76 | 100.0 | 148 | 100.0 |
| Women | 71 | 98.6 | 67 | 87.0 | 138 | 92.6 |
| Age, y | 31.9 (10.4) | | 38.4 (12.3) | | 35.3 (11.9) | |
| **All pre-adolescents** | 72 | 100.0 | 77 | 100.0 | 149 | 100.0 |
| Girls | 39 | 54.2 | 39 | 50.6 | 78 | 52.3 |
| Age, y | 11.1 (1.5) | | 11.1 (1.6) | | 11.1 (1.5) | |
| Income, ksh/month | | | | | | |
| < 10,000 ksh | 27 | 37.5 | 1 | 1.3 | 28 | 18.8 |
| 10,000–30,999 ksh | 37 | 51.4 | 17 | 22.1 | 54 | 36.2 |
| 31,000–50,999 ksh | 2 | 2.8 | 16 | 20.8 | 18 | 12.1 |
| >51,000 ksh | 3 | 4.2 | 41 | 53.2 | 44 | 29.5 |
| Education | | | | | | |
| None | 2 | 2.8 | 1 | 1.3 | 3 | 2.0 |
| Incomplete primary | 14 | 19.4 | 4 | 5.2 | 18 | 12.1 |
| Primary | 26 | 36.1 | 5 | 6.5 | 31 | 20.8 |
| Incomplete secondary | 14 | 19.4 | 7 | 9.1 | 21 | 14.1 |
| Secondary | 12 | 16.7 | 15 | 19.5 | 27 | 18.1 |
| Tertiary | 4 | 5.6 | 45 | 58.4 | 49 | 32.9 |
| Occupation | | | | | | |
| Unemployed* | 17 | 23.6 | 30 | 39.0 | 47 | 31.5 |
| Employed | 20 | 27.8 | 19 | 24.7 | 39 | 26.2 |
| Casual | 21 | 29.2 | 4 | 5.2 | 25 | 16.8 |
| Business | 10 | 13.9 | 22 | 28.6 | 32 | 21.5 |
| Unknown | 4 | 5.6 | 2 | 9.5 | 6 | 4.0 |
| Marital status | | | | | | |
| Married or cohabiting | 55 | 76.4 | 53 | 68.8 | 108 | 72.5 |
| Single | 10 | 13.9 | 17 | 22.1 | 27 | 18.1 |
| Divorced or separated | 5 | 6.9 | 2 | 2.6 | 7 | 4.7 |
| Widow | 2 | 2.8 | 5 | 6.5 | 7 | 4.7 |

* Includes students and housewives.

approximately five. In total, the biggest income group was participants earning 10,000–30,999 Ksh (about 77–237€) per month. Most participants residing in Embakasi belonged to this or lower income groups whereas majority of participants residing in Langata belonged to the highest income group (>51,000 Ksh per month), as expected. Most of the participants had completed primary or higher education. Most of those who had completed only primary education resided in Embakasi, whereas most of those who had completed tertiary education resided in Langata. The biggest difference occupation-wise was that casual jobs (employment on a need basis and the service can be terminated without notice), were more common in Embakasi. Marital status was similar between the two study areas.

The ownership of different indicators (assets) by the Wealth Index fifths showed that car, microwave, and refrigerator ownership was very clearly concentrated to those in the highest

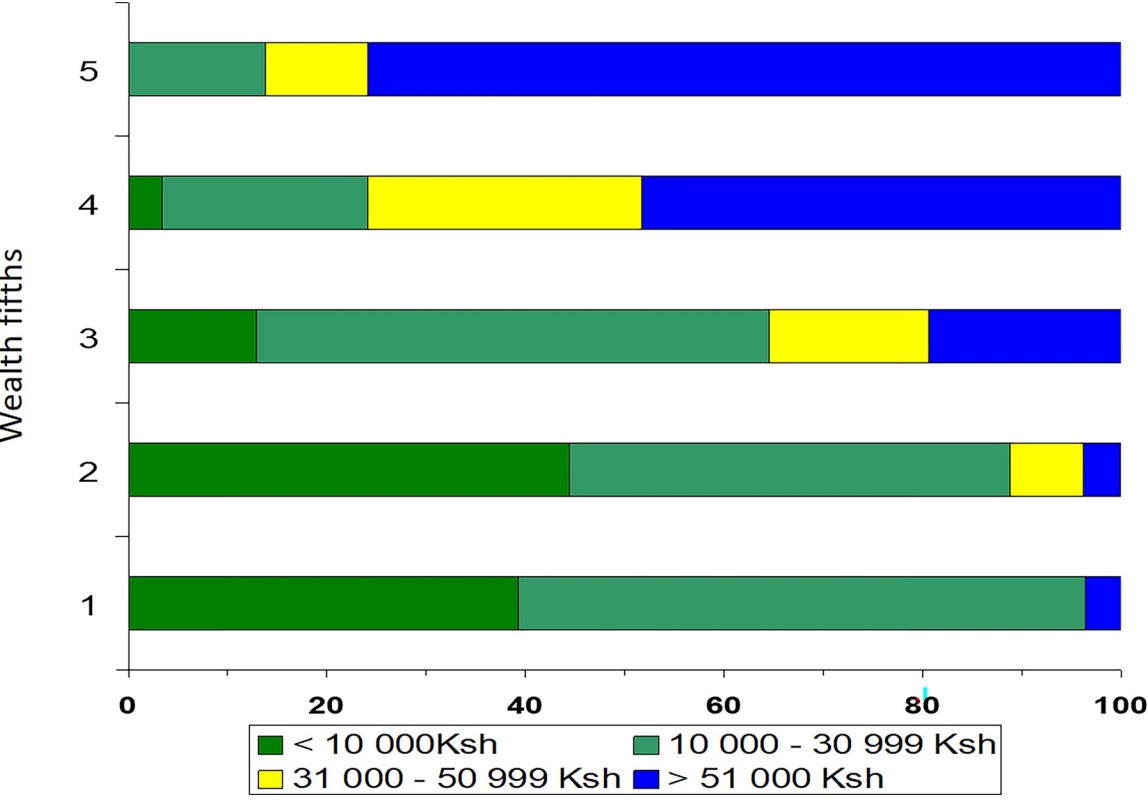

**Fig 1. Distribution of income classes among individuals in fifths of wealth index, among 149 households from two sub-counties in Nairobi County, Kenya.** Ksh = Kenyan shillings.

fifth of wealth, while no one in the lowest fifth reported having any of these assets. Kerosene stove was the only asset with evident concentration on the lowest fifth of wealth. By visual inspection, the most even distribution through wealth classes was observed for electric or gas stove, chair and television.

Comparison of the Wealth Index and simple self-reported income classes suggest a strong relationship. Almost all with <10,000 Ksh/month income (about 77 euro) belonged to the lowest 40% of Wealth Index (Fig 1). The distribution of families into the Wealth Index fifths in Embakasi and Langata is shown in Fig 2. As expected, in Embakasi, none of the participants belonged to the highest Wealth Index fifth whereas in Langata none of the participants belonged to the lowest Wealth Index.

The anthropometric results are shown in Table 2. The prevalence of pre-adolescents with overweight and obesity was significantly higher in Langata. About 30% of the pre-adolescent participants in Langata were classified as having at least overweight, whereas the respective number in Embakasi was only 6%. In striking contrast to the pre-adolescents, the prevalence of adults with overweight and obesity was high (65%) and very similar in the two study areas. It should be noted that the adult results represent mostly mothers, whereas the sample for pre-adolescents was quite evenly distributed between girls and boys. The mean BMI in the 11 participating adult men was 24.6 (SD 3.5) and in women (n = 138) 28.5 (SD 6.1), suggesting even without a statistical verification a clear sex difference. We assessed potential undernourishment using mid upper-arm circumference (MUAC). There was a small difference between the two sub-counties, and this was apparently explained by higher number of pre-adolescents with mild undernourishment in Embakasi (Table 2).

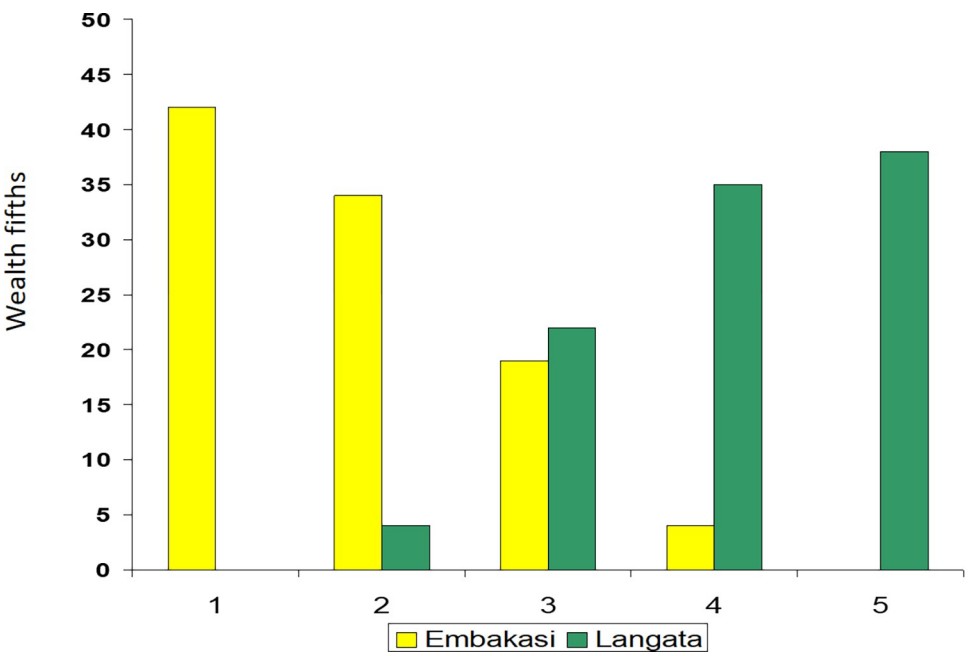

**Fig 2. Distribution of the wealth index (as fifths) in participants from Embakasi and Langata study areas, Nairobi County, Kenya.**

In a linear regression model, adjusted for age and sex, the beta-coefficient for Wealth Index was 0.01 (SE 0.0; p = 0.003) and for income 0.29 (SE 0.11; p = 0.009). The same trend was also seen when using a logistic regression model to explore associations with Wealth Index / income and overweight (the highest or two highest classes had an OR statistically significantly different from reference (S1 Table), but not with obesity. The main problem in using income classes in the logistic regression was related to the very low number of cases in the two highest income classes and this led to spuriously high odds ratios. In contrast to the pre-adolescents, the analysis of wealth/income vs. obesity did not show any evident associations among adults (S1 Table).

## Discussion

The main result in our study was that pre-adolescent overweight was highly prevalent in a middle-income area in Nairobi County, while the proportion of women with overweight or obesity was high in both the low- and middle-income areas. One of the characteristics of the lifestyle transition in lower middle-income economies is that the higher SES dominance in NCDs and NCD-related behaviour start to disappear [21,22]. Based on these findings, our results showing a high prevalence of obesity and NCD-related lifestyles in quite low SES areas in adult women in urban Nairobi was according to our hypothesis. It is good to realize that although the two areas were very different from a SES viewpoint, they do not represent the poorest or richest areas in Nairobi.

This study is unique in that it targeted pre-adolescents (9–14 years old), a group on which there is limited information on nutrition in Kenya and yet this group contributes to a fairly large proportion of the population. Our findings provide important new data on obesity, as well as on determinants of NCD-related lifestyles and risk factors in urban Nairobi City County, although the rather limited number of households prevents us from an accurate estimation of prevalence. The comparison of low- and middle-income mother-adolescent pairs is

**Table 2. Anthropometric measures and prevalence of underweight, overweight and obesity in pre-adolescents and adults (majority of the adults were women).**

| Pre- adolescents | Total | | Embakasi | | Langata | | |
|---|---|---|---|---|---|---|---|
| | Mean | SD | Mean | SD | Mean | SD | P[1] |
| BMI, z-score | -0.08 | 1.50 | -0.43 | 1.25 | 0.24 | 1.64 | 0.007 |
| MUAC, cm | 20.8 | 3.6 | 20.0 | 2.7 | 21.6 | 4.1 | 0.005 |
| | n | % | n | % | n | % | |
| Overweight (z-score >1.0) | 26 | 17.7 | 4 | 5.7 | 22 | 29.3 | <0.001 |
| Obesity (z-score >2.0) | 11 | 7.5 | 3 | 4.3 | 8 | 10.7 | 0.16 |
| Underweight (z-score < -2.0) | 10 | 6.8 | 4 | 5.7 | 6 | 8 | 0.61 |
| MUAC < 18.5 cm | 32 | 21.8 | 22 | 31.4 | 10 | 13.3 | 0.008 |
| MUAC < 16.0 cm | 7 | 4.8 | 4 | 5.7 | 3 | 4 | 0.62 |
| Adults | | | | | | | |
| | Mean | SD | Mean | SD | Mean | SD | P |
| BMI, kg/m2 | 28.2 | 6.1 | 27.6 | 5.8 | 28.8 | 6.3 | 0.24 |
| Waist circumference, cm | 91.2 | 14.4 | 89.5 | 14.1 | 92.8 | 14.5 | 0.16 |
| MUAC, cm | 32.2 | 5.4 | 31.7 | 5.0 | 32.6 | 5.7 | 0.31 |
| | n | % | n | % | n | % | |
| Overweight, ($>24.9$ kg/m$^2$) | 95 | 65.5 | 46 | 64.8 | 49 | 66.2 | 0.49 |
| Obesity, ($>29.9$ kg/m$^2$) | 49 | 33.8 | 23 | 32.4 | 26 | 35.1 | 0.43 |
| Central obesity (women >88 cm, men >102 cm) | 79 | 55.6 | 38 | 53.5 | 41 | 57.7 | 0.42 |
| Underweight ($<18.5$ kg/m$^2$) | 3 | 2.1 | 1 | 1.4 | 2 | 2.7 | 0.83 |
| MUAC women <22cm, men <23cm | 1 | 0.7 | 0 | 0 | 1 | 1.4 | NA[2] |

BMI = body mass index, MUAC = mid upper-arm circumference.

[1] Difference in anthropometric measurements between study areas was analyzed using analysis of variance for continuous variables, and with logistic regression for binary variables. Analyses were adjusted of participants' sex and age.

[2] There were too few cases to run the logistic regression for adult MUAC.

Both wealth (quintile classification) and income (class) predicted higher BMI z-score in adolescents (S1 Table).

novel in African context, and our data give a unique insight into urban residential areas with varying SES. These data are urgently needed to understand determinants of health and to plan health promotion interventions and programs. It is anticipated that the overall results of the present proposal can be extrapolated to other sub-Saharan African countries facing similar challenges and with comparable socioeconomic profiles and trajectories.

Comparison of the prevalence of pre-adolescent overweight and obesity between different studies is difficult, particularly because of different age-groups and similarity of the study areas. The prevalence of pre-adolescents with overweight/obesity in our sample had similar level, compared with findings from in Malawi and Benin, more than in Ghana and Cameroon, but less than in Djibouti and South Africa [23–26]. Out of these countries, Malawi is still a low-income economy, while Benin has just recently become a low-middle income economy, like Kenya. Also, Ghana and Cameroon are low-middle income economies, while South-Africa is a higher middle-income economy [3]. From this viewpoint, the prevalence of pre-adolescents with overweight and obesity in our sample was according to the expectations.

The critical question in our study, and around Sub-Saharan Africa, is whether overweight and obesity have become an issue also in the poorest segment of population [27]. We could not show this in our adolescent population, since the prevalence of both overweight and obesity was much higher in Langata, than in Embakasi. A similar SES-difference for younger children has been shown in South Africa [24] and Cameroon [26], but we are not aware of any studies using the same age-group as we did. Moreover, it should be noted that we did not

compare high and low SES, but middle and low SES. We hypothesize that due to the activities of daily living, the adolescents in Langata are less physically active because of using motorized transport to school and other activities than those in Embakasi whom the majority walk to school. Further research should be conducted to investigate the similarity in the level of overweight and obesity among the women in low and middles SES in this study.

In contrast to children, we found a high and similar prevalence of mothers with overweight and or obesity in both study areas. Already more than 10 years ago, Ziraba et al. [28] showed that the proportion with obesity increased more rapidly in Sub-Saharan countries among those with lower education. The very few participating men had clearly lower BMI, compared to women, but the small number prevents any interpretations. However, it seems that cultural expectations and beliefs may contribute to the high level of overweight and obesity among women in many Sub-Saharan countries [29]. Hence, in terms of preventive health policies, women in Sub-Saharan urban areas may be particularly responsive to negative effects of the lifestyle transition, regardless of their living conditions and household wealth.

The strengths of this study include two very well selected sub-counties, which enabled us to study socio-economically very different areas, but still within the low-middle range of SES. We chose the areas and potential households purposefully, but the final sample was invited by using a random selection of households. Most of the families invited participated in the study. This gives us a reason to expect that the families that fulfilled our study criteria and were included do not differ from those who were not included. Hence, we believe they represented the sub-county despite a rather small number. We used several methods to assess both nutritional status and the economic situation of the households. Comparison of the Wealth Index and self-reported income class indicated very similar site-differences and associations with indicators of over- and under-nourishment. Since the Wealth Index is quite cumbersome to calculate, self-reported income may work almost equally well in lower middle-income economies, at least in urban settings. However, the classification should be much denser than we had, and should be planned to describe particularly the lower range better.

The main weakness in this study was a limited number of households, although we met the initial power calculation for the expected difference of BMI z-score [14]. The power for identifying dichotomous (e.g., overweight vs. normal weight) was not as good. We were not able to study a larger number of households, since the total number of assessments done with each household was large and in total took for two visits 4—5h per household. Still, the analyses done with the entire sample (n = 149) gave similar results, compared to those using site-specific analyses, giving confidence that our conclusions are valid even for the sub-county level with this sample size. Moreover, in the linear regression model, the beta-coefficient for Wealth Index against BMI was very small (0.01), albeit significant. Hence, the interpretation should be done with caution.

The study was carried out among households with pre-adolescents between 9–14 years of age and their guardians in Nairobi City County and, thus, the research findings can only be specifically generalised to areas with similar characteristics and pre-adolescents of the same age group. This is a cross-sectional study, and the survey was done during one season (April-June 2019) only. Effects of seasonality and the direction of observed associations between variables cannot be assessed, and naturally any evidence of causality cannot be assumed.

In conclusion, we studied two Nairobian sub-counties, Embakasi (low SES) and Langata (middle SES). The thorough assessment with Wealth Index and income levels showed remarkable differences in the household situations between these two sites. The prevalence of pre-adolescents with overweight and obesity was clearly higher in Embakasi, but no difference was seen among guardians (mostly mothers). Moreover, the number of guardians (93% were mothers) with overweight was high in both sites. Neither of the studies sub-counties represents

true high-income areas. We can conclude that in Nairobi, pre-adolescent overweight was already highly prevalent in middle-income areas, while the proportion of women with overweight/obesity was high. These results suggest that a lifestyle promoting obesity is prevalent even in lower income areas in urban Kenya, and this should further be used as a strong justification for promoting healthy lifestyles across all SES classes.

## Supporting information

**S1 Table. Association between adolescent anthropometrics and wealth index.**
(XLSX)

**S1 Text. A detailed description of design, methods, and practical experiences.**
(DOC)

**S2 Text. Additional figures and tables related to the construction of the wealth index.**
(DOC)

**S3 Text. Inclusivity in global research.**
(DOCX)

## Author Contributions

**Conceptualization:** Noora Kanerva, Lucy Joy Wachira, George E. Owino, Esther L. Anono, Maijaliisa Erkkola, Nils Swindell, Gareth Stratton, Vincent Onywera, Mikael Fogelholm.

**Data curation:** George E. Owino, Mikael Fogelholm.

**Formal analysis:** Noora Kanerva, George E. Owino, Maijaliisa Erkkola.

**Funding acquisition:** Mikael Fogelholm.

**Investigation:** Sophie Ochola, Noora Kanerva, George E. Owino, Esther L. Anono, Hanna M. Walsh, Nils Swindell, Vincent Onywera, Mikael Fogelholm.

**Methodology:** Noora Kanerva, Lucy Joy Wachira, George E. Owino, Esther L. Anono, Hanna M. Walsh, Maijaliisa Erkkola, Nils Swindell, Gareth Stratton, Vincent Onywera, Mikael Fogelholm.

**Project administration:** Lucy Joy Wachira, Victor Okoth.

**Resources:** Hanna M. Walsh, Vincent Onywera, Mikael Fogelholm.

**Supervision:** Sophie Ochola, Noora Kanerva, Lucy Joy Wachira, George E. Owino, Esther L. Anono, Victor Okoth, Maijaliisa Erkkola, Nils Swindell, Vincent Onywera, Mikael Fogelholm.

**Validation:** Sophie Ochola, Noora Kanerva, Lucy Joy Wachira, George E. Owino, Esther L. Anono, Hanna M. Walsh, Maijaliisa Erkkola, Nils Swindell, Vincent Onywera, Mikael Fogelholm.

**Visualization:** Maijaliisa Erkkola, Nils Swindell, Gareth Stratton.

**Writing – original draft:** Sophie Ochola.

**Writing – review & editing:** Noora Kanerva, Lucy Joy Wachira, George E. Owino, Esther L. Anono, Hanna M. Walsh, Victor Okoth, Maijaliisa Erkkola, Nils Swindell, Gareth Stratton, Vincent Onywera, Mikael Fogelholm.

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
