## [Decision Letter · Decision Letter 0]

9 Jun 2022

PGPH-D-22-00151

WEALTH AND OBESITY IN PRE-ADOLESCENTS AND THEIR GUARDIANS: A FIRST STEP IN EXPLAINING NON-COMMUNICABLE DISEASE-RELATED BEHAVIOUR IN TWO AREAS OF NAIROBI CITY COUNTY

Dear Dr. Ochola,

Thank you for submitting your manuscript to PLOS Global Public Health. After careful consideration, we feel that it has merit but does not fully meet PLOS Global Public Health’s publication criteria as it currently stands. Therefore, we invite you to submit a revised version of the manuscript that addresses the points raised during the review process.

Please submit your revised manuscript by . If you will need more time than this to complete your revisions, please reply to this message or contact the journal office at globalpubhealth@plos.org. Please include the following items when submitting your revised manuscript:

We look forward to receiving your revised manuscript.

Kind regards,

Catherine Elizabeth Draper

Academic Editor

Journal Requirements:

1. Please include a complete copy of PLOS’ questionnaire on inclusivity in global research in your revised manuscript. Our policy for research in this area aims to improve transparency in the reporting of research performed outside of researchers’ own country or community. The policy applies to researchers who have travelled to a different country to conduct research, research with Indigenous populations or their lands, and research on cultural artefacts. The questionnaire can also be requested at the journal’s discretion for any other submissions, even if these conditions are not met.  Please find more information on the policy and a link to download a blank copy of the questionnaire here: https://journals.plos.org/plosone/s/best-practices-in-research-reporting. Please upload a completed version of your questionnaire as Supporting Information when you resubmit your manuscript.”

2. We ask that a manuscript source file is provided at Revision. Please upload your manuscript file as a .doc, .docx, .rtf or if latex provide both .pdf and .tex.

3. Please amend your detailed Financial Disclosure statement. This is published with the article. It must therefore be completed in full sentences and contain the exact wording you wish to be published.

a. Please clarify all sources of funding (financial or material support) for your study. List the grants (with grant number) or organizations (with url) that supported your study, including funding received from your institution. 

b. State the initials, alongside each funding source, of each author to receive each grant.

c. State what role the funders took in the study. If the funders had no role in your study, please state: “The funders had no role in study design, data collection and analysis, decision to publish, or preparation of the manuscript.”

4. Please update the 'Competing Interests' statement with this "The authors have declared that no competing interests exist". 

5. Please provide separate figure files in .tif or .eps format.

6. We have noticed that you have uploaded Supporting Information files, but you have not included a list of legends. Please add a full list of legends for your Supporting Information files after the references list. 

7. In the online submission form, you indicated that "THE DATA IS AVAILABLE BY REQUEST FROM HELSINKI UNIVERSITY FINLAND OR KENYATTA UNIVERSITY, NAIROBI KENYA". All PLOS journals now require all data underlying the findings described in their manuscript to be freely available to other researchers, either 1. In a public repository, 2. Within the manuscript itself, or 3. Uploaded as supplementary information.

Additional Editor Comments (if provided):

Reviewers' comments:

Reviewer's Responses to Questions

**Comments to the Author**

1. Does this manuscript meet PLOS Global Public Health’s publication criteria? Is the manuscript technically sound, and do the data support the conclusions? The manuscript must describe methodologically and ethically rigorous research with conclusions that are appropriately drawn based on the data presented.

Reviewer #1: Yes

2. Has the statistical analysis been performed appropriately and rigorously?

Reviewer #1: Yes

3. Have the authors made all data underlying the findings in their manuscript fully available (please refer to the Data Availability Statement at the start of the manuscript PDF file)?

Reviewer #1: Yes

4. Is the manuscript presented in an intelligible fashion and written in standard English?

Reviewer #1: Yes

5. Review Comments to the Author

Reviewer #1: This study examines associations between wealth, income, and bmi z-scores in pre-adolescents and found that higher wealth and income predicted higher bmi z-scores. The study's hypotheses are grounded in the literature and are in line with other studies in similar countries.

Several times throughout the paper, the phrase "results suggest a rapid transition in lifestyle...". I suggest avoiding the word "transition" because it sounds like the study examined changes in patterns over time instead of associations. I think the authors should reword this.

Lines 60-61 ("The overall aim of this study..."): Remove the word "interactions". This led me to believe the study examined statistical interactions, but it did not.

Lines 62-64: Do not include the purpose of the qualitative portion here. It sounds like this paper includes the qualitative results. Consider including this information in the "future directions" paragraph in the discussion.

Lines 148-150: Do not need to include how interviews were conducted.

Lines 177: Include version of R software.

Lines 189-196: Move this information to the measures section. Information about the measure does not belong in the statistical analysis section.

Statistical analysis section: Clarify how your linear and logistic regression models were set up (i.e., outcome variable, predictors, covariates).

Results - please provide a table displaying all regression model results.

Results - include if there were significant demographic differences between included and excluded participants

Results - a beta of .01 is very small. This should be acknowledged as a limitation in the discussion and its practical significance should be discussed.

Lines 290-291: Provide statistics to support this sentence.

6. PLOS authors have the option to publish the peer review history of their article (what does this mean?). If published, this will include your full peer review and any attached files.

**Do you want your identity to be public for this peer review?** For information about this choice, including consent withdrawal, please see our Privacy Policy.

Reviewer #1: No

---

## [Decision Letter · Decision Letter 1]

7 Dec 2022

PGPH-D-22-00151R1

WEALTH AND OBESITY IN PRE-ADOLESCENTS AND THEIR GUARDIANS: A FIRST STEP IN EXPLAINING NON-COMMUNICABLE DISEASE-RELATED BEHAVIOUR IN TWO AREAS OF NAIROBI CITY COUNTY

Dear Dr. Ochola,

Thank you for submitting your manuscript to PLOS Global Public Health. After careful consideration, we feel that it has merit but does not fully meet PLOS Global Public Health’s publication criteria as it currently stands. Therefore, we invite you to submit a revised version of the manuscript that addresses the points raised during the review process.

We look forward to receiving your revised manuscript.

Kind regards,

Catherine Draper

Academic Editor

Journal Requirements:

1. We noticed that you used "not shown" in the manuscript. We do not allow these references, as the PLOS data access policy requires that all data be either published with the manuscript or made available in a publicly accessible database. Please amend the supplementary material to include the referenced data or remove the references.

Additional Editor Comments (if provided):

The reviewer had some further comments that need addressing before the manuscript can be considered for publication.

Reviewers' comments:

Reviewer's Responses to Questions

**Comments to the Author**

1. If the authors have adequately addressed your comments raised in a previous round of review and you feel that this manuscript is now acceptable for publication, you may indicate that here to bypass the “Comments to the Author” section, enter your conflict of interest statement in the “Confidential to Editor” section, and submit your "Accept" recommendation.

Reviewer #1: (No Response)

2. Does this manuscript meet PLOS Global Public Health’s publication criteria? Is the manuscript technically sound, and do the data support the conclusions? The manuscript must describe methodologically and ethically rigorous research with conclusions that are appropriately drawn based on the data presented.

Reviewer #1: Yes

3. Has the statistical analysis been performed appropriately and rigorously?

Reviewer #1: Yes

4. Have the authors made all data underlying the findings in their manuscript fully available (please refer to the Data Availability Statement at the start of the manuscript PDF file)?

Reviewer #1: Yes

5. Is the manuscript presented in an intelligible fashion and written in standard English?

Reviewer #1: Yes

6. Review Comments to the Author

Reviewer #1: The lines listed in the response to review did not match the manuscript, so it was difficult to find the changes that were made.

Lines 60-61: The word "interactions" was not removed. I suggest changing the purpose statement to something like: "The overall aim of this study was to examine indicators of wealth and obesity (i.e., poverty, education, gender, and place of residence) in association with physical (in)activity, dietary quality and variability, and overweight/obesity among guardians and pre-adolescent children in low- and middle-income areas in urban Nairobi.

Lines 62-62: The part describing the qualitative study was not removed as stated in the response to review.

Lines 78: There is only one parenthesis in this sentence: "In this paper, we report methods and results from the socio-demographic indices, including wealth and anthropometric assessments)."

Lines 137: Consider renaming this section. Instead of "Background data" maybe call it something like "Socioeconomic data", "Wealth indicator data" or "Survey data". "Background data" is very vague and I wasn't sure what it was referring to.

Lines 149-154: Thank you for moving information about the Wealth Index to this paragraph. I think it fits well in this section.

Lines 181-190: Thank you for including more information on how the linear and logistic regression models were set up. Could you also include how each binary variable was coded? For example, overweight/obese=1; not overweight/obese=0?

Lines 290-292: Consider removing the sentence: "As could be expected from the comparison of the two sub-counties, both wealth (quintile classification) and income (class) predicted higher BMI z-score in adolescents." A sentiment like this should be in the discussion where you state what you expected or did not expect to find. I suggest only reporting the results in this section.

Line 316: Consider removing "e.g.," from this sentence so that it reads: "...although the rather limited number of households prevents us from an accurate estimation of prevalence."

Line 314: Consider removing "It is expected that the..." from the beginning of the sentence in line 314 so that the sentence reads: "The findings of this research will provide novel and important new data on obesity..."

Lines 359: I think you are referring to "sub-counties" not "sub-countries".

Lines 304-310 (first paragraph of discussion): Consider changing this paragraph to explain the overall takeaways from the paper before discussing specific findings.

Lines 324-329: This information should go in the limitations/weaknesses paragraph.

Discussion: I suggest adding a paragraph on discussing the finding that wealth and income predicted higher BMI z-scores in pre-adolescents. This is one of the most important findings.

7. PLOS authors have the option to publish the peer review history of their article (what does this mean?). If published, this will include your full peer review and any attached files.

**Do you want your identity to be public for this peer review?** For information about this choice, including consent withdrawal, please see our Privacy Policy.

Reviewer #1: No

---

## [Editor Report · Decision Letter 2]

2 Feb 2023

WEALTH AND OBESITY IN PRE-ADOLESCENTS AND THEIR GUARDIANS: A FIRST STEP IN EXPLAINING NON-COMMUNICABLE DISEASE-RELATED BEHAVIOUR IN TWO AREAS OF NAIROBI CITY COUNTY

PGPH-D-22-00151R2

Dear Professor Ochola,

We are pleased to inform you that your manuscript 'WEALTH AND OBESITY IN PRE-ADOLESCENTS AND THEIR GUARDIANS: A FIRST STEP IN EXPLAINING NON-COMMUNICABLE DISEASE-RELATED BEHAVIOUR IN TWO AREAS OF NAIROBI CITY COUNTY' has been provisionally accepted for publication in PLOS Global Public Health.

Best regards,

Catherine Elizabeth Draper

Academic Editor
